# Systematic Microbiome Dysbiosis Is Associated with IgA Nephropathy

Fengtao Cai,[a,b] Chenfen Zhou,[c] Na Jiao,[d] Xinling Liang,[b] Zhiming Ye,[b] Wei Chen,[e] Qiongqiong Yang,[f] Hui Peng,[g] Ying Tang,[h] Chaoqun Niu,[c] Guoping Zhao,[c,i] Zefeng Wang,[c,j] Guoqing Zhang,[c] [ID]Xueqing Yu[b,a,k]

[a]School of Medicine, South China University of Technology, Guangzhou, China

[b]Department of Nephrology, Guangdong Provincial People's Hospital (Guangdong Academy of Medical Sciences), Southern Medical University, Guangzhou, China

[c]National Genomics Data Center & Bio-Med Big Data Center, CAS Key Laboratory of Computational Biology, Shanghai Institute of Nutrition and Health, University of Chinese Academy of Sciences, Chinese Academy of Sciences, Shanghai, China

[d]National Clinical Research Center for Child Health, the Children's Hospital, Zhejiang University School of Medicine, Hangzhou, Zhejiang, China

[e]Department of Nephrology, The First Affiliated Hospital, Sun Yat-sen University, Guangzhou, China

[f]Department of Nephrology, Sun Yat-sen Memorial Hospital, Sun Yat-sen University, Guangzhou, China

[g]Department of Nephrology, The Third Affiliated Hospital of Sun Yat-sen University, Guangzhou, China

[h]Department of Nephrology, The Third Affiliated Hospital of Southern Medical University, Guangzhou, China

[i]Hangzhou Institute for Advanced Study, University of Chinese Academy of Sciences, Hangzhou, China

[j]CAS Center for Excellence in Molecular Cell Science, Chinese Academy of Sciences, Shanghai, China

[k]Guangdong-Hong Kong Joint Laboratory on Immunological and Genetic Kidney Diseases, Guangzhou, China

Fengtao Cai, Chenfen Zhou, and Na Jiao contributed equally to this work. Author order was determined in order of the contribution to this study.

**ABSTRACT** IgA nephropathy (IgAN) is reportedly associated with microbial dysbiosis. However, the microbiome dysregulation of IgAN patients across multiple niches remains unclear. To gain a systematic understanding of microbial dysbiosis, we conducted large-scale 16S rRNA gene sequencing in IgAN patients and healthy volunteers across 1,732 oral, pharynx, gut, and urine samples. We observed a niche-specific increase of several opportunistic pathogens, including *Bergeyella* and *Capnocytophaga* in the oral and pharynx, whereas some beneficial commensals decreased in IgAN patients. Similar alterations were also observed in the early versus advanced stage of chronic kidney disease (CKD) progression. Moreover, *Bergeyella, Capnocytophaga*, and *Comamonas* in the oral and pharynx were positively associated with creatinine and urea, indicating renal lesions. Random forest classifiers were developed by using the microbial abundance to predict IgAN, achieving an optimal accuracy of 0.879 in the discovery phase and 0.780 in the validation phase.

**IMPORTANCE** This study provides microbial profiles of IgAN across multiple niches and underlines the potential of these biomarkers as promising, noninvasive tools with which to differentiate IgAN patients for clinical applications.

**KEYWORDS** IgA nephropathy, oral, pharynx, gut, urine, microbiome

Immunoglobulin A nephropathy (IgAN) is the most common primary glomerulonephritis worldwide and is a leading cause of kidney failure, specifically in the Asian population (1, 2). Approximately 20 to 40% of IgAN patients progress to end-stage kidney disease within 10 to 20 years after the initial biopsy (3, 4). Patients with IgAN typically have mucosal infections involving the upper respiratory, gastrointestinal, and urinary tracts before or during the onset of IgAN (5, 6). Moreover, mucosal infection would lead to abnormal mucosal immunity and mediate the production of IgA and galactose-deficient IgA1 (Gd-IgA1) (7, 8). However, the exact antigen-causing mucosal infection remains poorly defined in IgAN patients.

**Ad Hoc Peer Reviewer** [ID] Ramon Garcia Maset, University College London

Address correspondence to Xueqing Yu, yuxueqing@gdph.org.cn.

The authors declare no conflict of interest.

Multiple lines of evidence suggest that microbiota and their derived metabolites exert a crucial impact on mucosal infection, immune homeostasis (9–11), and microbiome dysbiosis in IgAN. A previous analysis using the saliva samples of 31 Chinese patients and 30 controls showed that the relative abundances of *Capnocytophaga* and *SR1_genera_incertae_sedis* were enriched in IgAN patients and that *Haemophilus* was positively associated with serum IgA (12). In tonsils, patients with IgAN had higher proportions of *Rahnella*, *Ruminococcus_g2*, and *Clostridium_g21*, according to one study of 21 patients and 23 controls in a Korean population (13). Many studies on the gut microbial composition in IgAN patients have suggested an increase in some pathogenic genera, such as *Escherichia-Shigella* (14–18) and *Eggerthella* (15), with a reduction in some beneficial bacteria, such as *Bifidobacterium* and *Blautia* spp. (16). Additionally, it was reported that pharyngitis was often associated with the acute onset of IgAN (19), and the urine itself was involved with many clinical indicators for the evaluation of IgAN. However, most of the previous studies were carried out with a small number of patients (usually less than 100), with no microbiota profile on the pharynx and urine of IgAN patients being characterized. Renal biopsy is the gold standard for IgAN diagnosis; however, its underlying disadvantages, such as injuries, pain, fever, and perirenal hematoma, sometimes make it infeasible for patients (20). Hence, we seek to examine whether microbial biomarkers could serve as a noninvasive complementary approach to diagnose IgAN.

In this study, we recruited a total of 505 participants and performed 16S rRNA gene sequencing on their oral, pharynx, gut, and urine samples. In the discovery cohort of 372 participants, we analyzed the IgAN-related multiniche microbiota compositions, coabundance networks, and associations between microbial genera and clinical indices, and we developed classifiers for detecting IgAN patients, based on microbial markers. The performance of those classifiers was further assessed with an independent cohort of 133 individuals. Moreover, we characterized the multiniche microbiome in the progression of IgAN, based on chronic kidney disease (CKD) classification. These findings could present new insights regarding the diagnosis, prevention, and treatment of IgAN.

## RESULTS

**Study design and clinical characteristics.** In total, 1,732 samples from 505 qualified individuals were collected in two separate batches during our study. In the first batch, the samples of four body niches (oral, pharynx, gut, and urine) in 245 IgAN patients and 127 healthy controls (HCs) were collected as the discovery cohort. Accordingly, the same samples from 65 IgAN patients and 68 controls served as the validation cohort (Fig. 1). In both the discovery cohort and the validation cohort, there were no statistically significant differences in age, gender, or body mass index (BMI) between IgAN patients and HCs ($P > 0.05$). The patients were divided into early (CKD stages 1 to 2) and advanced (CKD stages 3 to 5) groups, based on the clinical stages of CKD (Table 1). The samples of early and advanced patients were also shown in Fig. 1.

As a reference, we first examined the typical clinical indicators used in IgAN diagnosis and found significant differences in creatinine, urea, the estimated glomerular filtration rate (eGFR), and Gd-IgA1 between the IgAN group and the HCs ($P < 0.05$). The clinical characteristics of the early and advanced patients in the discovery and validation cohorts are shown in Table S1.

**Microbiome dysbiosis across four niches in IgAN patients.** To explore the microbiota changes of IgAN, we performed 16S rRNA gene sequencing on four niches (oral, pharynx, gut, urine), and we observed shared (around 30%) and unique amplicon sequence variants (ASVs) between IgAN patients and HCs in the discovery cohort (Fig. S1). Then, altered microbial alpha diversity values, as estimated by several common indices, were observed across multiple niches (Table S2). For instance, the Shannon index was increased significantly in the oral cavity (Kruskal-Wallis test, $P = 0.0198$) but was decreased significantly in the gut ($P = 0.0034$) of IgAN patients, compared to the respective controls (Fig. 2A). However, there were no significant differences of alpha

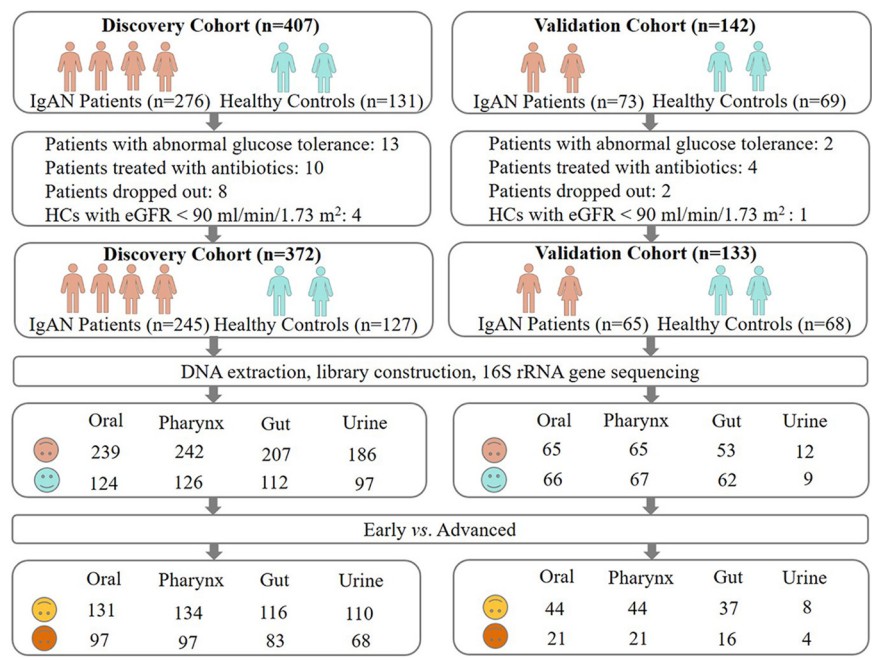

**FIG 1** Study design. 549 participants were initially enrolled in our study. After filtering with strict inclusion and exclusion criteria, a total of 505 eligible participants were included and further divided into two cohorts, according to the different sequencing batches. Oral swabs, pharynx swabs, feces, and urine samples were obtained from each participant and detected via 16S rRNA gene sequencing. Moreover, the IgAN patients were further divided into early and advanced groups, according to the different CKD stages.

diversity in the pharynx and urine samples ($P > 0.05$) (Table S2; Fig. 2A). The microbial beta diversity, based on the Bray-Curtis distance, varied significantly not only between disease status (PERMANOVA, $P = 0.004$) (Fig. 2B) but also across body niches ($P = 0.001$) (Fig. 2B).

Next, we observed distinct microbiota profiles across different niches at the phylum level. The oral and pharynx microbiome was dominated by *Firmicutes*, and this was followed by *Proteobacteria*, *Bacteroidota*, *Actinobacteriota*, and *Fusobacteriota* in both HCs and IgAN patients. The dominant phyla of the gut microbiota compositions were similar to those of the oral and pharynx samples, whereas the top five phyla of the urine microbiota were different from those of the above niches, ranking as *Proteobacteria*,

**TABLE 1** Clinical characteristics of participants in the discovery and validation cohorts[a]

| Clinical indices | Discovery cohort (n = 372) | | | Validation cohort (n = 133) | | |
|---|---|---|---|---|---|---|
| | IgAN (n = 245) | HCs (n = 127) | P value | IgAN (n = 65) | HCs (n = 68) | P value |
| Age (yrs) | 38.2 ± 9.94 | 36.71 ± 11.39 | 0.13 | 38.06 ± 9.2 | 37.18 ± 10.96 | 0.27 |
| Gender (female) | 132 (53.88%) | 55 (43.31%) | 0.07 | 34 (52.31%) | 40 (58.82%) | 0.56 |
| BMI (kg/m²) | 22.45 ± 3.17 | 22.85 ± 3.74 | 0.28 | 21.6 9 ± 5.71 | 22.53 ± 3.22 | 0.95 |
| Creatinine (umol/L) | 155.72 ± 183.77 | 69.24 ± 12.71 | 1.59E-21 | 112.34 ± 59 | 68.67 ± 16.57 | 4.55E-09 |
| Urea (mmol/L) | 7.8 ± 5.48 | 4.64 ± 1.15 | 9.01E-15 | 6.79 ± 2.41 | 4.86 ± 1.19 | 1.40E-08 |
| eGFR (mL/min/1.73m²) | 68.56 ± 33.73 | 110.48 ± 13.84 | 7.15E-27 | 77.5 ± 30.87 | 108.51 ± 24.41 | 5.69E-08 |
| Gd-IgA1/IgA1 | 0.84 ± 0.53 | 0.55 ± 0.3 | 3.49E-09 | 0.79 ± 0.49 | 0.65 ± 0.32 | 0.22 |
| IgA1 (ug/mL) | 830.16 ± 373.83 | 785.19 ± 283.23 | 0.74 | 849.9 ± 303.16 | 652.54 ± 301.04 | 4.84E-04 |
| Gd-IgA1 (ug/mL) | 6.12 ± 3.65 | 4.06 ± 2.25 | 3.14E-11 | 5.96 ± 2.96 | 3.72 ± 1.71 | 2.82E-06 |
| | | | | | | |
| CKD stages | | | | | | |
| Stages 1 to 2 | 136 (55.51%) | | | 44 (67.69%) | | |
| Stages 3 to 5 | 98 (40%) | | | 21 (32.31%) | | |

[a]Continuous variables were expressed as means ± standard deviations and compared using Wilcoxon rank-sum tests. Categorical variables were expressed as percentages and compared using Chi-square tests. IgAN, immunoglobulin A nephropathy; HCs, healthy controls; BMI, body mass index; eGFR, estimated glomerular filtration rate; Gd-IgA1, galactose-deficient immunoglobulin A 1; CKD, chronic kidney disease.

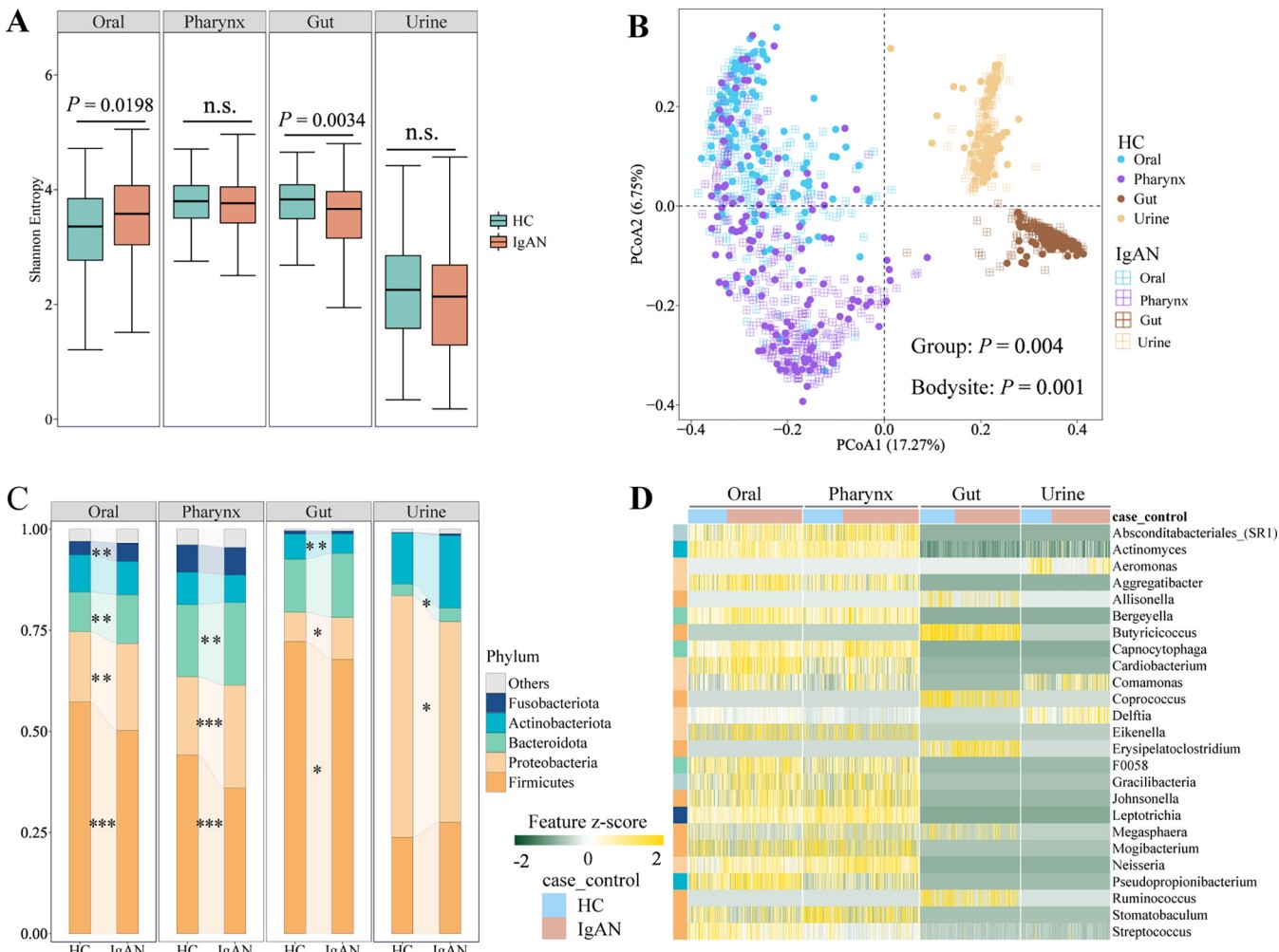

**FIG 2** Microbiome dysbiosis across four body niches in IgAN. (A) The alpha diversity of each body niche was estimated by Shannon entropy for IgAN versus HCs. *P* values were calculated via Kruskal-Wallis tests. (B) The beta diversity was estimated using the Bray-Curtis distance for IgAN versus HCs. *P* values were calculated via PERMANOVA. (C) The proportions of relative abundance at the phylum level in IgAN patients and HCs across four body niches. (D) Differential microbial genera from each niche for IgAN versus HCs. Differential microbial phyla and genera were identified via Wilcoxon rank-sum tests, and *P* values were further adjusted via the FDR method in R. The heat map showed the results of the FDR *P* < 0.05, where the feature *z* score was calculated on the log value of the relative abundance (to avoid infinite values from the logarithm, a pseudocount of 1E−06 was added to all of the values). n.s., not significant; *, FDR *P* < 0.2; **, FDR *P* < 0.05; ***, FDR *P* < 0.01; HC, healthy control; IgAN, immunoglobulin A nephropathy; PCoA, principal coordinates analysis; PERMANOVA, permutational multivariate analysis of variance.

*Firmicutes*, *Actinobacteriota*, *Bacteroidota*, and *Fusobacteriota* (Fig. 2C). These results highlighted the niche-specific microbial compositions in both IgAN patients and HCs. In particular, the predominant phylum *Firmicutes* significantly decreased in the oral cavity, pharynx, and gut of the IgAN patients, in contrast to the HCs, while showing no distinct difference in the urine samples. The other dominant phylum, namely, *Proteobacteria*, was significantly increased in the oral, pharynx, and gut samples and was significantly reduced in the urine samples of the IgAN patients (all FDR *P* < 0.2) (Fig. 2C).

Furthermore, we compared the taxonomic compositions at the genus level and found a significant microbial imbalance in all niches of IgAN patients, and these were mainly characterized by significant decreases in some beneficial bacteria and increases in some opportunistic bacteria. In the oral samples, a total of 29 microbial genera were significantly altered between the IgAN patients and the HCs (FDR *P* < 0.2) (Fig. S2). Among these, short-chain fatty acid (SCFA)-producing bacteria, including *Megasphaera* and *Stomatobaculum*, decreased. Meanwhile, some pathogens or opportunistic pathogens were significantly enriched, such as *Bergeyella*, *Capnocytophaga*, *Comamonas*, *Neisseria*, *Cardiobacterium*, and

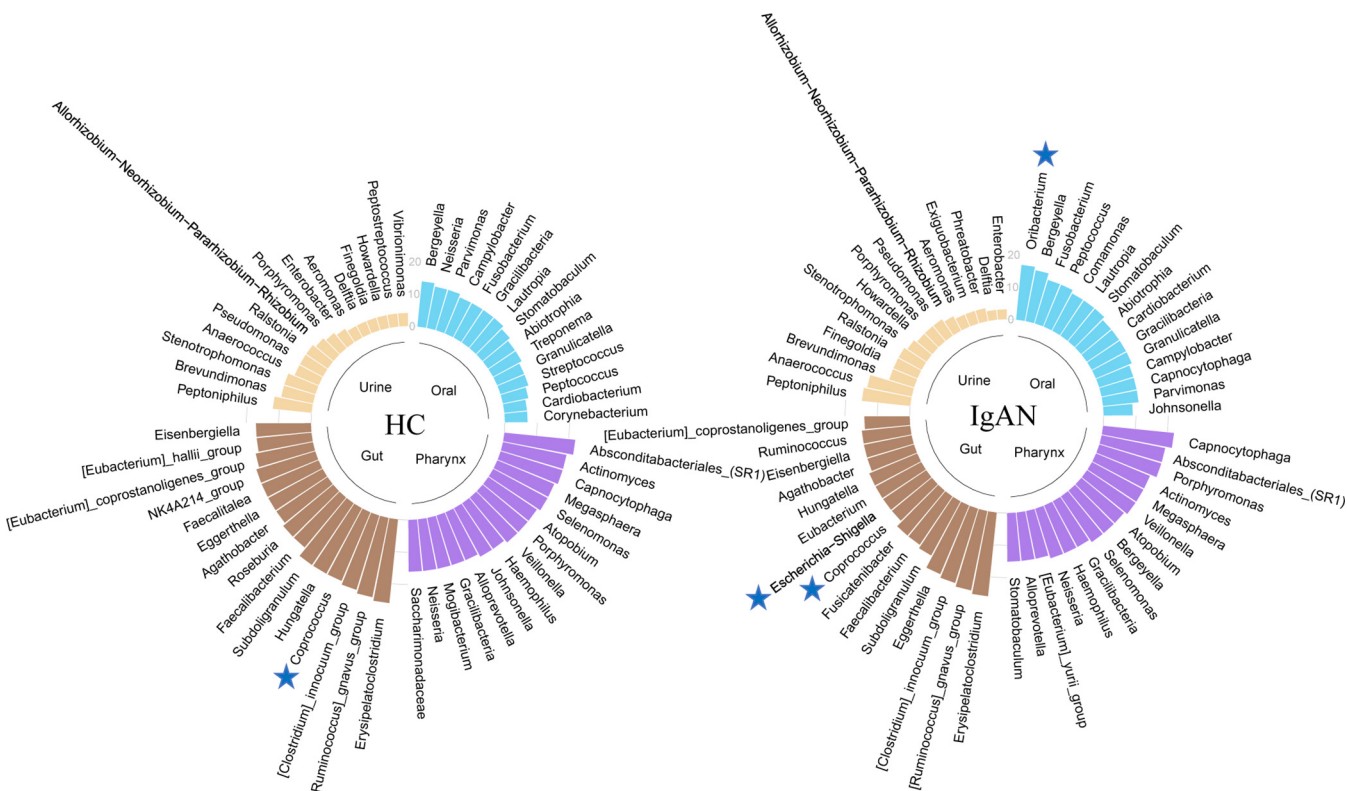

**FIG 3** The hub microbes with the most associations in the coabundance networks. The microbial genera were ranked based on their associations (node degree) in each coabundance network, and only the top 15 genera with the highest node degree were identified as hub genera. The blue pentagram marks the genera with obvious degree changes. The height of each bar represents the size of the node degree, and the colors represent the different body niches (blue, oral; purple, pharynx; Earth yellow, gut; faint yellow, urine).

*Eikenella* (FDR $P < 0.05$) (Fig. 2D). In the pharynx, there were 31 genera with relative abundance changes in the IgAN patients (FDR $P < 0.2$) (Fig. S2). Notably, 10 genera with stringent statistical significance were shown in the heat map (FDR $P < 0.05$) (Fig. 2D), including *Actinomyces*, *Delftia*, *Mogibacterium*, *Stomatobaculum*, and *Streptococcus*, the relative abundance of which declined. However, *Bergeyella*, *Capnocytophaga*, *Comamonas*, *Gracilibacteria*, and *Neisseria* were more abundant in patients. Regarding gut flora, 35 genera significantly changed between the IgAN patients and HCs (FDR $P < 0.2$) (Fig. S2), where the histamine-producing *Allisonella* and the SCFA-producing *Butyricicoccus*, *Coprococcus*, *Ruminococcus* were significantly decreased in patients, whereas the pathogen *Erysipelatoclostridium* was dramatically increased (FDR $P < 0.05$) (Fig. 2D). However, relatively few genera with significant differences were identified in the urinary microbiome (FDR $P < 0.2$) (Fig. S2), and only *Aeromonas*, with significantly reduced abundance, was reported in the IgAN patients (FDR $P < 0.05$) (Fig. 2D).

**Microbial coabundance associations across multiple niches in IgAN patients and controls.** To uncover the potential interactions among bacteria, we constructed coabundance networks, based on the differential microbial genera (FDR $P < 0.2$) (Fig. S2), and we identified hub microbes, according to the node degree of each genus. The results showed that these microbial associations and hub genera varied not only in different disease statuses but also in niches, suggesting that changes in both the relative abundance and interactions of microbiota may affect the progression of IgAN. In detail, the oral ecological network of the IgAN patients (133 edges and 29 nodes) was relatively more connected than was that of the HCs (113 edges and 28 nodes) (Fig. 3; Fig. S3A), whereas the hub genera *Oribacterium* in the IgAN group had more significant correlations with pathogens, including *Bergeyella* and *Capnocytophaga*, and had been reported to be a possible biomarker in liver cancer (21). In the pharynx, the coabundance network of the IgAN patients (207 edges and 31 nodes) and the hub genera distribution were similar to those of the HCs

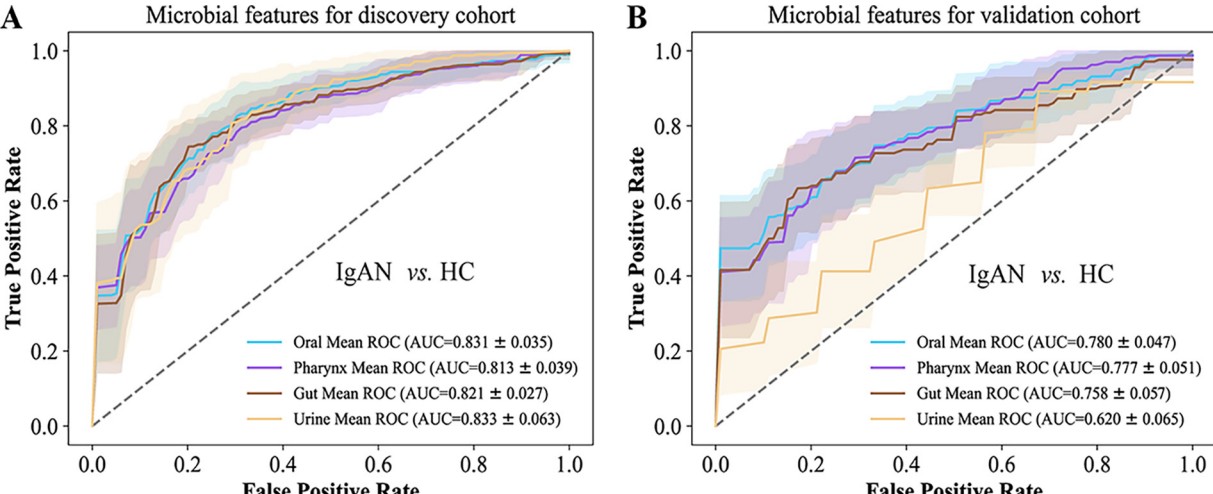

**FIG 4** Microbial classifiers constructed via a random forest algorithm for IgAN across four niches. (A and B) Models for distinguishing IgAN patients from HCs with microbial markers in the discovery cohort and validation cohort, respectively. The ROC curve and 95% CI shaded area were shown in the same color for each body niche (blue, oral; purple, pharynx; Earth yellow, gut; faint yellow, urine). ROC, receiver operating characteristic; AUC, area under curve; CI, confidence interval.

(216 edges and 31 nodes) (Fig. 3; Fig. S3B). In the gut, the patients' network (209 edges and 33 nodes) was less complex than that of the controls (245 edges and 34 nodes), and the hub genera also varied. For instance, the associations of the pathogen *Escherichia-Shigella* were more complicated, whereas the SCFA-producing bacteria *Coprococcus, Bifidobacterium,* and *Butyricicoccus* showed a trend towards less connectivity in the IgAN patients (Fig. 3; Fig. S3C). Moreover, the urine ecological networks were more sparse than those of other niches, probably due to the low microbial diversity (Fig. 2A), and these were similar between the IgAN patients (55 edges and 19 nodes) and the HCs (52 edges and 18 nodes) (Fig. 3; Fig. S3D).

**Correlations between differential microbial genera and clinical indicators of IgAN.** To explore the possible influence of microbes on renal function, we calculated the Spearman rank correlations between clinical parameters and the differential microbial genera from each niche in the discovery cohort (abs [correlation coefficient] ≥ 0.3, FDR $P$ < 0.1). The results showed that these microbial genera were closely related to urea, eGFR, creatinine, complement C3, and IgA, among which the microbiota in the oral and pharynx samples, such as *Bergeyella, Capnocytophaga, Comamonas,* and *Gracilibacteria*, were positively correlated with urea and creatinine and were negatively correlated with eGFR. In particular, the pharyngeal opportunistic pathogen *Bergeyella* was positively correlated with the size of the multifocal atrophic area of the renal tubular epithelium (MAARTE), which is indicative of renal reabsorption capacity. In the urine samples, *Aeromonas* was positively correlated with eGFR but was negatively correlated with creatinine, and the pathogen *Enterobacter* was positively correlated with the IgA from blood. However, the gut bacteria were not found to be significantly associated with these clinical indicators (Fig. S4A).

**Identification and validation of microbial classifiers for IgAN.** To assess the capability of microbial biomarkers to distinguish IgAN patients from HCs, we performed model construction analyses, based on microbes from multiple niches. As expected, the microbial biomarkers from each niche achieved good performance in distinguishing IgAN patients. In detail, the model built with 10 differential bacteria genera in the oral samples showed an accuracy of 0.831, which consisted of *Capnocytophaga, Bergeyella, Comamonas, Stomatobaculum, Pseudopropionibacterium, Oribacterium, Megasphaera, Gracilibacteria, Johnsonella,* and *Cardiobacterium*, according to the Gini importance (Table S3). The analogous models were constructed using the respective differential bacteria genera in the pharynx, gut, and urine (Table S3), and these achieved similar accuracies of 0.813, 0.821, and 0.833, respectively (Fig. 4A). The upper respiratory tract microbes, namely, *Comamonas, Stomatobaculum, Capnocytophaga, Bergeyella,* and *Gracilibacteria*, were selected as predictive features in both

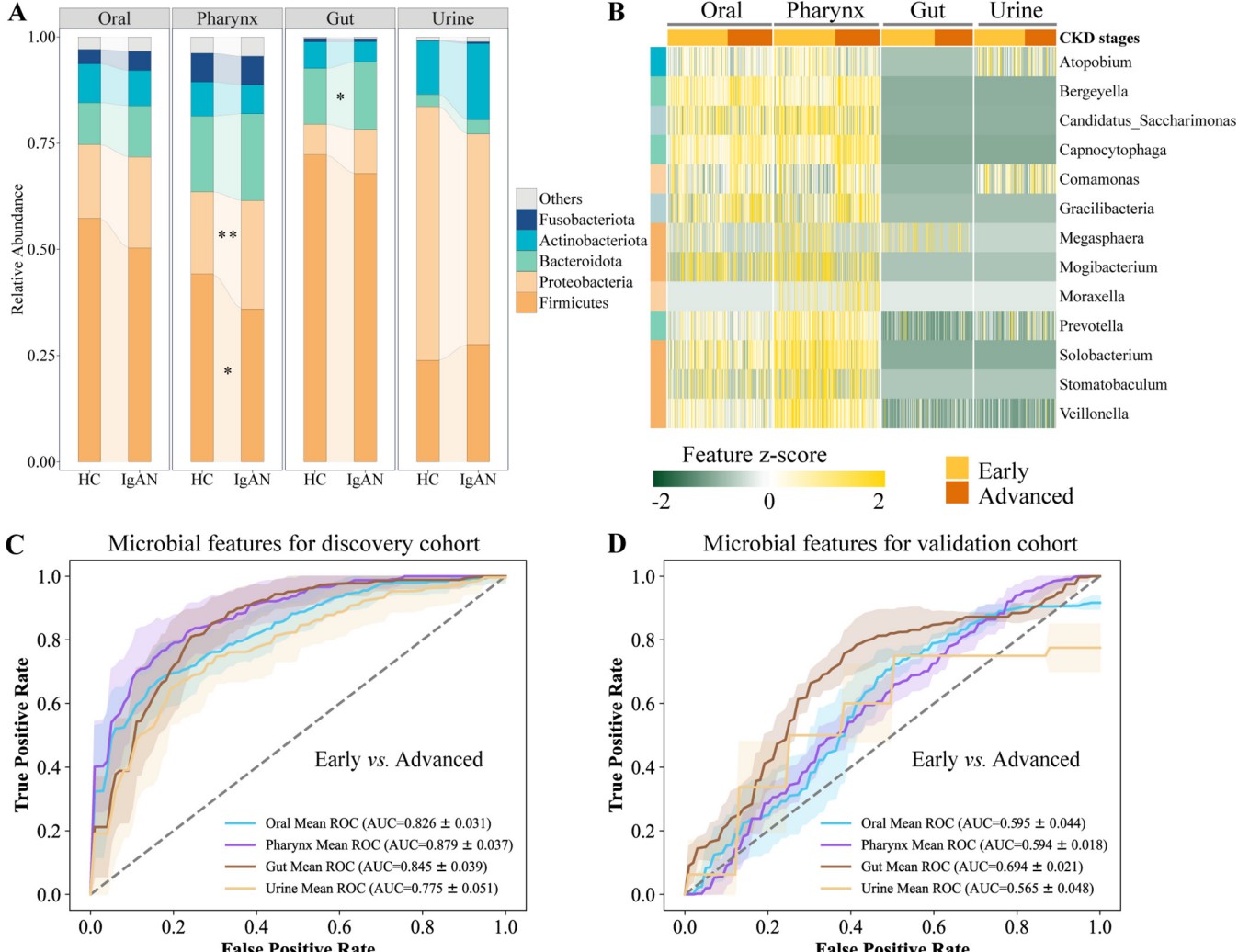

**FIG 5** Microbial alterations and random forest classifiers in the early and advanced stages of IgAN patients across four niches. (A) Proportions of relative abundance at the phylum level in the early and advanced stages of IgAN patients across four body niches. (B) Differential microbial genera from each niche for early versus advanced patients. Differential microbial phyla and genera were identified via Wilcoxon rank-sum tests, and the *P* values were further adjusted using the FDR method in R. The heat map showed the results of the FDR *P* < 0.05, where the feature *z* score was calculated on the log value of the relative abundance (to avoid infinite values from the logarithm, a pseudocount of 1E−06 was added to all of the values). *, FDR *P* < 0.2; **, FDR *P* < 0.05. (C and D) Models for distinguishing early patients from advanced patients with microbial markers in the discovery cohort and validation cohort, respectively. The ROC curve and 95% CI shaded area were shown in the same color for each body niche (blue, oral; purple, pharynx; Earth yellow, gut; faint yellow, urine). ROC, receiver operating characteristic; AUC, area under curve; CI, confidence interval.

the oral and pharynx models, suggesting their potential to detect IgAN. Next, we used an independent external cohort to validate these biomarkers and obtained the accuracies of 0.780, 0.777, 0.758, and 0.620 for the oral, pharynx, gut, and urine models, respectively (Fig. 4B). These results indicated that the microbial markers of each niche could differentiate IgAN patients from HCs, and the performances in the validation cohort were acceptable, except for those of the urine samples.

**Microbiome dysbiosis across four niches in the early and advanced stages of IgAN patients.** Furthermore, we wondered whether microbes were altered in the development of IgAN, and similar analyses were performed on early (CKD stages 1 to 2) and advanced (CKD stages 3 to 5) stages of IgAN patients in the discovery cohort. Among the pharynx microbiota, the phylum *Firmicutes* was significantly decreased, whereas *Proteobacteria* were significantly increased in the advanced IgAN patients, compared to the early IgAN patients (FDR *P* < 0.2) (Fig. 5A). Additionally, 20 bacteria genera of the oral cavity changed significantly between the early and advanced patients (FDR *P* < 0.2) (Fig. S5), and *Bergeyella*, *Capnocytophaga*, *Comamonas*, and *Gracilibacteria* were more

abundant in the advanced patients (FDR $P < 0.05$) (Fig. 5B). In the pharynx, 31 genera were significantly altered between the early group and the advanced group (FDR $P < 0.2$) (Fig. S5), with 13 genera being markedly varied (FDR $P < 0.05$) (Fig. 5B), including a reduction of the SCFA-producing bacteria *Candidatus_Saccharimonas* and *Megasphaera* and the lactate-producing bacteria *Veillonella* as well as an increase of *Bergeyella*, *Capnocytophaga*, *Comamonas*, *Gracilibacteria*, and *Moraxella*, in the advanced stage of IgAN. However, our data only reported nine gut genera and seven urine genera with significant abundance differences between the early and advanced stages of IgAN patients (FDR $P < 0.2$) (Fig. S5). These results imply that the microbiome disturbance of the upper respiratory tract (oral and pharynx) may be critical in the development of IgAN.

The differential microbes of CKD stages showed significant correlations with IgAN-related clinical indices. In agreement with the results of the IgAN-HCs group, *Bergeyella*, *Capnocytophaga*, *Comamonas*, and *Gracilibacteria* in the oral cavity and pharynx also showed positive correlations with creatinine but negative correlations with eGFR. Other pharyngeal microbiota displayed more clinical relevance, compared to the results in the IgAN-HCs group. For example, the proinflammatory genus *Mogibacterium* and the lactate-producing genus *Veillonella* were negatively related to creatinine and urea, which were not observed in the IgAN-HCs group. In the pharynx, the opportunistic pathogen *Neisseria* had a positive correlation with albumin/creatinine (mg/g), and the SCFA producer *Stomatobaculum* showed a positive correlation with eGFR. Overall, we speculate that the microbes in the oral and pharynx of IgAN patients are more closely related to renal function. This is consistent with the phenomenon of obvious inflammation in the pharynxes of IgAN patients in clinical practice (Fig. S4B).

We also investigated the performance of microbial signatures across multiple niches to distinguish early patients from advanced patients, based on a random forest algorithm. The respective accuracy of the oral, pharynx, gut, and urine models were 0.826, 0.879, 0.845, and 0.775 in the discovery cohort (Fig. 5C), which was comprised of some identical bacteria, compared to the IgAN-HCs group, such as *Capnocytophaga*, *Bergeyella*, *Comamonas*, *Gracilibacteria* in the oral and pharynx samples, *Allisonella* in the gut samples, as well as *Aeromonas*, *Delftia*, and *Vibrionimonas* in the urine samples (Tables S3 and S4), thereby indicating the potential roles of these bacteria in the occurrence and development of IgAN. However, the accuracies of the oral, pharynx, gut, and urine models were relatively low in the validation cohort (Fig. 5D).

## DISCUSSION

Microbiota dysregulation has recently been recognized as one of the major pathogenic factors of IgAN, in which the abnormal communications between the host immune system and the mucosal microbiota may be related to the occurrence and development of IgAN (22–24). However, the exact profiles of the multiniche microbiota and specific bacteria for IgAN are unclear. We recruited 505 Chinese participants and performed 16S rRNA gene sequencing on 1,732 oral, pharynx, gut, and urine samples. Next, we examined the multiniche microbiome profiles, coabundance networks, and associations between the microbes and the clinical indices of IgAN in the discovery cohort. Moreover, we identified microbiome biomarkers for distinguishing IgAN patients versus HCs as well as early versus advanced IgAN patients in the discovery cohort, and these were further verified in an independent cohort. These findings collectively indicated that microbial markers would be a promising tool for the detection and molecular classification of IgAN.

Previous studies on the IgAN microbiome usually focused on oral or gut microbiota (12, 14–18). Our study provided the first systematic survey of microbial diversity across four distinct niches of IgAN patients, revealing niche-specific patterns of microbial dysbiosis. For example, we found a higher oral alpha diversity in IgAN patients by using oral swabs, which was different from the lower alpha diversity tendency ($P > 0.05$) in oral saliva that was observed previously with a smaller cohort (12). Consistent with the results of previous studies, the gut alpha diversity was decreased in IgAN patients (15, 25). The beta diversity was found to have significant differences across the four niches

between patients and controls, suggesting the particularity of microbes in IgAN. Furthermore, the genus-level comparison revealed that 94 genera were significantly altered in total across four niches, with 54 genera being increased in the IgAN patients (FDR $P < 0.2$). The most increased microbes are pathogens or opportunistic pathogens, such as *Capnocytophaga* in the oral and pharynx and *Escherichia-Shigella* in the gut. Conversely, the SCFA-producing bacteria, such as *Megasphaera* in the oral and pharynx and *Bifidobacterium* in the gut, were all significantly decreased in the IgAN patients. These findings are supported by the results of previous studies that *Capnocytophaga* being more present within the saliva of IgAN patients (11) and that *Capnocytophaga ochracea* could produce an immunosuppressive factor and degrade immunoglobulins (26, 27). Additionally, it is reported that *Escherichia-Shigella* is characterized by the invasion and inflammatory destruction of the human colonic epithelium (28), and the intraperitoneal administration of *Escherichia coli* could induce the glomerular deposition of IgA and C3 in mice (29). In particular, we found that the changes in the microbiota in the upper respiratory tract (oral and pharynx) were more apparent than were those of the gut and urine samples under a much more stringent cutoff (FDR $P < 0.05$). This is consistent with the clinical manifestation that IgAN patients always develop oropharyngeal infections (30).

Our study revealed a close correlation between various microbes in the upper respiratory tract and renal lesions in IgAN. Four genera (*Capnocytophaga*, *Comamonas*, *Bergeyella*, and *Gracilibacteria*) in the upper respiratory tract were found to be negatively correlated with eGFR (correlation coefficient $< -0.3$, FDR $P < 0.01$), suggesting that these microbes may have a detrimental effect on kidney function in IgAN. In the pharynx, we found a positive correlation of *Bergeyella* with the renal tubular atrophy (correlation coefficient $> 0.3$, FDR $P < 0.01$), which is an important indicator of disease severity. Moreover, *Mogibacterium* and *Veillonella* in the pharynx were positively correlated with complement C3 (correlation coefficient $> 0.3$, FDR $P < 0.01$), which may reflect the inflammatory state of the patients and relate to IgA production. Furthermore, *Capnocytophaga* and *Comamonas* in the upper respiratory tract showed a tendency of positive correlations with serum Gd-IgA1 (correlation coefficient $> 0.2$, FDR $P < 0.05$), and their overproduction has been described as an indispensable pathogenic factor in IgAN (5, 23, 31, 32). These results suggest that the microbes of the upper respiratory tract may affect renal function in IgAN and that the local treatment of oropharyngitis may become a potential treatment for IgAN patients.

Currently, the only confirmed method for IgAN diagnosis is kidney biopsy, which is an invasive procedure that is harmful to many patients (20). Therefore, discovering noninvasive biomarkers for IgAN diagnosis and prediction, such as microbial markers, has become an urgent need. Microbial markers are regarded as a potential noninvasive tool for the diagnosis of diverse diseases, such as type 2 diabetes (33), liver cirrhosis (34), and colorectal cancer (35, 36). Accordingly, pathogenic associations between gut microbiota and kidney diseases (the gut-kidney axis) have been widely investigated (37). Previously the gut microbiome dysbiosis has been analyzed in nondialysis chronic kidney disease (CKD), resulting in five optimal microbial markers for CKD classification (38). A recent study also observed gut microbiome imbalances in IgAN patients and utilized seven bacteria to construct a random forest model (18). However, to the best of our knowledge, there were no studies on the microbial markers of the other three niches in IgAN. Our microbiome signatures of the oral, pharynx, gut, and urine have successfully distinguished the IgAN patients with a predictive accuracy of >0.8 in all niches. These biomarkers were further verified in an external cohort, suggesting the applicability of microbial markers to distinguish other IgAN populations.

Our study also emphasizes that microbial markers can be used to distinguish IgAN patients in different stages. The pharynx microbiome model performed well in distinguishing early from advanced IgAN patients with an accuracy of 0.879 in the discovery cohort. However, the microbial signatures had limited predictive power in the validation cohort, probably due to the small sample size and imbalance of the external cohort (Fig. 1). Taken altogether, our results suggest that microbiome markers could

differentiate the IgAN patients from the controls, from patients in the early stage to the advanced stage of IgAN, and that microbiome markers could possibly be used for the diagnosis and treatment of IgAN. However, our results were only implemented in the Chinese population, and the main conclusions should be verified in other populations from different regions with larger sample sizes in the future.

In this study, we have systematically delineated the first microbiome profiles of IgAN across multiple niches in a relatively large cohort and found the potential biomarkers to distinguish patients from controls. Our work also revealed significant correlations between clinical indices and microbes in the upper respiratory tract of IgAN. Although more clinical validations and mechanism investigations are needed, our study emphasized the importance of understanding the pathogenesis, diagnosis, prevention, and treatment of IgAN.

## MATERIALS AND METHODS

**Patient recruitment.** Between March of 2021 and October of 2022, 276 IgAN patients and 131 healthy volunteers from Guangdong Provincial People's Hospital, The First Affiliated Hospital of Sun Yat-sen University, Sun Yat-sen Memorial Hospital of Sun Yat-sen University, The Third Affiliated Hospital of Sun Yat-sen University, and The Third Affiliated Hospital of Southern Medical University were recruited. 31 IgAN patients with an abnormal glucose tolerance, who were treated with antibiotics, or who dropped out were excluded from the study. 4 healthy controls (HCs) with an eGFR of less than 90 mL/min/1.73 $m^2$ were also excluded (Fig. 1). Finally, 245 IgAN patients and 127 HCs served as the discovery cohort. Between November of 2020 and February of 2021, 73 IgAN patients and 69 HCs from Guangdong Provincial People's Hospital were recruited. Similarly, eight IgAN patients and one HC were excluded (Fig. 1). The remaining 65 IgAN patients and 68 HCs were considered to be the validation cohort.

The details of the inclusion criteria were as follows: participants between 14 to 70 years old and with a normal BMI (18.5 to 27 kg/$m^2$) were included. The diagnosis of IgAN was to be confirmed by a kidney biopsy with immunofluorescence studies for IgA deposits. Patients with pregnancy, inflammatory bowel disease, irritable bowel syndrome, other intestinal diseases, renal replacement therapies (hemodialysis, peritoneal dialysis, renal transplantation), secondary IgAN (systematic lupus erythematosus, rheumatic disease, IgA vasculitis), or type 2 diabetes mellitus were excluded. The HCs without renal, oral, respiratory, gastrointestinal, or urinary tract diseases were voluntarily recruited. Participants who were treated with antibiotics, microbial agents, traditional Chinese medicine within 4 weeks, glucocorticoid, or other immunosuppressants within 24 weeks before enrollment were excluded to minimize the confounding factors. This study complied with the Declaration of Helsinki and was approved by the Ethics Committee of Guangdong Provincial People's Hospital (GDREC2020234H [R2], Guangzhou, China). Written informed consent was obtained from all of the patients and HCs who were involved.

**Sample collection.** The oral samples were collected by gently swabbing over the mucosa on both sides of the mouth. The pharynx samples were obtained by swabbing between the tonsillar pillars and the tonsils on both sides of the throat. The plasma and urine samples were collected with fecal samples from each recruited volunteer at the same time point. The oral swabs, pharynx swabs, and fresh fecal samples were then quickly placed in liquid nitrogen boxes and transferred to the laboratory for subpackaging. After the subpackaging, the samples were transferred promptly to a −80℃ cryogenic refrigerator for freezing. Fresh urine samples were centrifuged for 10 min at 10,000 × $g$ (4℃) with the precipitation preservation at −80℃. 5 mL of blood were collected, and serum samples were obtained via centrifugation at 3,000 rpm for 10 min. The samples were then preserved at −80℃. The entire collection and packaging process was completed within 1 h.

**Clinical variables.** Details about the demographics and clinical data, such as age, gender, BMI (body mass index), 24 h urine protein excretion, serum immunoglobin, complement component 3, and creatinine, were recorded. The estimated glomerular filtration rate (eGFR) was calculated with the CKD-EPI equation (39). All renal biopsies were processed via microscopy (light, electron, immunofluorescence). We scored the renal biopsies according to the Oxford Classification in patients with IgAN. Other serological markers, including IgA1 and Gd-IgA1, were detected via enzyme-linked immunosorbent assay (ELISA) (Immuno-Biological Laboratories, Japan).

**DNA extraction and 16S rRNA gene sequencing.** Total microbial genomic DNA was extracted from oral swabs, pharynx swabs, fecal samples, and urine samples using the E.Z.N.A. soil DNA Kit (Omega Biotek, Norcross, GA, U.S.), according to the manufacturer's instructions. A NanoDrop spectrophotometer (Thermo Fisher Scientific, USA) and 1% agarose gel electrophoresis were used to monitor the concentration and size of the DNA, which could then be kept at −80℃ until further use. The V3-V4 hypervariable regions of the bacterial 16S rRNA gene were amplified using the primer pairs 338F (5′-ACTCCTACGG GAGGCAGCAG-3′) and 806R (5′-GGACTACHVGGGTWTCTAAT-3′) (40) by using an ABI GeneAmp 9700 PCR thermocycler (ABI, CA, USA). The PCR mixture included 4 $\mu$L of 5× FastPfu buffer, 2 $\mu$L of 2.5 mM dNTPs, 0.8 $\mu$L of each primer (5 $\mu$M), 0.4 $\mu$L of FastPfu polymerase, 10 ng of template DNA, and ddH$_2$O to a final volume of 20 $\mu$L. The PCR amplification cycling conditions were as follows: initial denaturation at 95℃ for 3 min, 27 cycles of denaturing at 95℃ for 30 s, annealing at 55℃ for 30 s, extension at 72℃ for 45 s, single extension at 72℃ for 10 min, and end at 4℃. All of the samples were amplified in triplicate. The polymerase chain reaction (PCR) product was extracted from 2% agarose gel, purified using an

AxyPrep DNA Gel Extraction Kit (Axygen Biosciences, Union City, CA, USA), according to the manufacturer's instructions, and quantified using a Quantus Fluorometer (Promega, USA).

The purified amplicons were pooled in equimolar amounts and paired-end sequenced on an Illumina MiSeq PE300 platform (Illumina, San Diego, USA), according to the standard protocols by Sinotech Genome Technology Co. (Shanghai, China).

**16S rRNA gene sequencing data processing.** The raw data from the 16S rRNA gene sequencing were analyzed using QIIME2 (v2020.11) (41). Briefly, the reads were denoised and filtered using divisive amplicon denoising algorithm 2 (DADA2) (42). The amplicon sequence variants (ASVs) were obtained with single nucleotide resolution. The ASVs were taxonomically classified and mapped to Silva-138-99 reference sequences by using the classify-sklearn plugin in QIIME2 (43). Then, the alpha diversity was computed using several common indexes, and the beta diversity was computed by using the Bray-Curtis distances among samples via the R package "vegan" (v2.5-7). The feature tables were converted to relative abundance tables. Moreover, the microbial genera that were present in at least 20% of the samples with a relative abundance of greater than 0.01% were selected for subsequent statistical analysis.

**Coabundance network analysis.** We applied the SparCC algorithm (44), which was robust for compositional data, to investigate the associations among differential microbiota at the genus level. Correlation coefficients were calculated with 10 iterations, and $P$ values were conducted from 1,000 bootstrap correlations for IgAN patients or HCs, which were further adjusted by the false discovery rate (FDR) method in R. Then, the correlation coefficients with absolute values of more than 0.2 and FDR $P$ values of less than 0.05 were selected for visualization in Gephi (v0.9.2) (45).

**Associations between differential microbial genera and clinical indices.** The correlations between the microbiota and the clinical indices were calculated via Spearman's rank correlation coefficient by using the R package Hmisc (v4.6-0). The correlation coefficients with absolute values of more than 0.3 and FDR $P$ values of less than 0.1 were selected in each body niche. The results of four body niches were recorded and visualized using the R package "pheatmap" (v1.0.12).

**Random forest modeling and validation.** Based on differential microbial genera, a random forest algorithm was used to construct classifiers to distinguish the IgAN patients from the HCs in the discovery cohort. At first, feature selection was performed using the "Boruta" (v7.0.0) package in R with costumed parameters (pValue = 0.05, mcAdj = T, maxRuns = 1000), which iteratively removed irrelevant features and retained final features to build a model. Next, undersampling was applied 20 times, as the number of IgAN patients was about twice the number of HCs. Then, the "caret" (v6.0-90) package was performed for hyperparameter tuning, including mtry, ntree, nodesize, and maxnodes. Finally, classification models were constructed against selected features, and the best combination of hyperparameters was chosen, using 5-fold cross-validation to avoid issues of overfitting. We verified the robustness of the microbial markers in an external cohort. Similar to the above steps, these identified microbial markers were used to construct 5-fold cross-validation models in the oral cavity, pharynx, and gut samples, respectively. The leave-one-out cross-validation (LOOCV) method was used for classifier construction in urine samples because of the small amount (Fig. 1).

Additionally, we also constructed the classifiers to distinguish early from advanced IgAN patients, using a similar method to that described above, except without undersampling, as the early and advanced patient sample sizes in the discovery cohort were relatively balanced (Fig. 1). The LOOCV method was also applied separately in the oral cavity, pharynx, gut, and urine samples of the validation cohort.

**Statistical analysis.** The continuous variables in the clinical indicators were statistically tested by using the Wilcoxon rank-sum test, and the categorical variables (such as gender) were statistically tested by using the Chi-square test. The Kruskal-Wallis test was used for statistical tests of alpha diversity, and PERMANOVA was performed for the beta diversity, where a $P$ value of <0.05 was defined as indicative of a statistically significant result. The Wilcoxon rank-sum test was also employed to identify the differentially abundant microbial features (phylum level and genus level) between HCs and IgAN patients as well as early and advanced IgAN patients, and in these analyses, the $P$ values were further adjusted by the FDR method. In addition, an FDR $P$ value of <0.2 was defined as indicative of a statistically significant result, and an FDR $P$ value of <0.05 was defined as indicative of a much more statistically significant result. MaAsLin2 (http://huttenhower.sph.harvard.edu/maaslin) was used for multivariable association testing between clinical characteristics (gender for gut samples and BMI for urine samples) and microbial features, using the default parameters (46). All of the statistical analyses described above were performed in R-4.1.2. The receiver operating characteristic (ROC) curves and the 95% CI shadow were visualized by using matplotlib (v3.5.1) in Python3.

**Data availability.** All of the 16S rRNA gene sequencing raw data can be viewed in NODE (http://www.biosino.org/node) by pasting the accession number (OEP003670) into the text search box or through the URL: https://www.biosino.org/node/project/detail/OEP003670.

## SUPPLEMENTAL MATERIAL

Supplemental material is available online only.
**SUPPLEMENTAL FILE 1**, PDF file, 1.2 MB.

## ACKNOWLEDGMENTS

We are grateful for all of the subjects who participated in this study.

F.C. collected the samples and drafted the manuscript. C.Z. analyzed the data and drafted the manuscript. N.J. analyzed the data and revised the manuscript. X.L., Z.Y.,

W.C., Q.Y., H.P., and Y.T. collected the samples. C.N. managed some metadata. G.Z. designed the project. Z.W. revised the manuscript. G.Z. and X.Y. designed the project and revised the manuscript. All authors read and approved the final manuscript.

This work was supported by grants from the National Natural Science Foundation of China (81920108008, 32070087), Guangdong-Hong Kong-Macao-Joint Labs Program from Guangdong Science and Technology (2019B121205005), and National Key R&D Program of China (2020YFC2005003, 2021YFF0703802).

We declare no competing interests.

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
