## [Reviewer comments · Microbiology Spectrum]

Microbiology Spectrum

Systematic microbiome dysbiosis are associated with IgA nephropathy

Fengtao cai, chenfen zhou, na jiao, xinling liang, zhiming ye, wei chen, qiongqiong yang, hui peng, ying tang, chaoqun niu, Guo-Ping Zhao, Zefeng Wang, guoqing zhang, and xueqing Yu

Corresponding Author(s): xueqing Yu, Guangdong Provincial People's Hospital

Review Timeline:

Submission Date:	December 18, 2022
Editorial Decision:	March 10, 2023
Revision Received:	April 21, 2023
Accepted:	May 4, 2023

Editor: Wei-Hua Chen

Reviewer(s): Disclosure of reviewer identity is with reference to reviewer comments included in decision letter(s). The following individuals involved in review of your submission have agreed to reveal their identity: Ramon Garcia Maset (Reviewer #1)

Transaction Report:

DOI: <https://doi.org/10.1128/spectrum.05202-22>

March 10, 2023

Prof. xueqing Yu
Guangdong Provincial People's Hospital
106th, Zhongshan Road II
Guangzhou
China

Re: Spectrum05202-22 (Systematic microbiome dysbiosis are associated with IgA nephropathy)

Dear Prof. xueqing Yu:

Thank you for submitting your manuscript to Microbiology Spectrum. Your manuscript has been evaluated by external experts. While their opinions were generally positive, they also raised substantial concerns. Attached please their comments. Thus, I would like to you to submit a revised version of your study to properly address their concerns.

Link Not Available

Sincerely,

Wei-Hua Chen

Journals Department
Reviewer comments:

Reviewer #1 (Comments for the Author):

- All the acronyms should be explained at least once in the main text.
- Alpha diversity was investigated by the authors using Shannon indexes. Some of the common alpha diversity indices are the following: (1) Chao Index, (2) Simpson Index, (3) Shannon Index, (4) ACE Index, and (5) Good's Coverage Index. Could the authors explain why they decided to use Shannon index to study alpha diversity and not the other of the indices? At least authors should mention why Shannon index was chosen.
- "Shannon index was increased in the oral cavity (Kruskal-Wallis test, $P = 0.0198$) but decreased in the gut ($P = 0.0034$)". Authors should indicate if the p-value is significant. Otherwise, it is confusing if the increase/decrease is significant or not.

- The authors should consider plotting the data using Venn diagram showed the shared and unique operational Taxonomic Unit (OTU) all samples from gut, urine, oral and pharynx. Furthermore, the authors should consider plotting the distribution of microbial community by using Circos at phylum level to compare gut, urine, oral and pharynx from IgAN patient vs control and early stage vs advanced.
- "In particular, the phylum Firmicutes was significantly decreased in oral cavity, pharynx, and gut of IgAN patients contrast to HCs, while there was no distinct difference in urine samples". The authors did not specify if the decrease is significant or not and a p-value was not given.
- "Furthermore, bacteria genera of nine in gut and seven in urine were changed obviously in the advanced stage of IgAN patients compared to early ones ($P < 0.05$, Figure S2). However, there were no significant differences characterized in gut and urine microbiome under much stricter filtering condition". These sentences does not make sense,, Could the authors clarify this?
- "After setting a stricter threshold ($P < 0.001$), the changes of microbiota in the oral and pharynx were much more obvious than that of gut and urine. The authors should clarify the threshold they use for the statistical analysis and when the threshold is statistically significant or not. It is hard to interpret the data.
- The authors studied the correlation of clinical parameters with the differential microbial genera, including eGFR, urea, and complement C3. In the following analysis of the identification and validation of microbial classifiers, the authors include haematuria to refine the model, being crucial. Could the authors use urinalysis data (level of blood, pH, the colour of the urine) to study their correlation with the differential microbial genera?
- Gender differences can be an essential factor not only in human research but also in preclinical research. IgAN Male patients presented with more severe clinical and pathological changes than female patients. The authors should consider analysing the data in gender categories. Clustering the data by gender might another important comparison as early-stage vs advanced stage.

Reviewer #3 (Comments for the Author):

please see the review attachment for my comments.

Staff Comments:

Preparing Revision Guidelines

Please return the manuscript within 60 days; if you cannot complete the modification within this time period, please contact me. If you do not wish to modify the manuscript and prefer to submit it to another journal, please notify me of your decision immediately so that the manuscript may be formally withdrawn from consideration by Microbiology Spectrum.

- All the acronyms should be explained at least once in the main text.
- Alpha diversity was investigated by the authors using Shannon indexes. Some of the common alpha diversity indices are the following: (1) Chao Index, (2) Simpson Index, (3) Shannon Index, (4) ACE Index, and (5) Good's Coverage Index. Could the authors explain why they decided to use Shannon index to study alpha diversity and not the other of the indices? At least authors should mentioned why Shannon index was chosen.
- "Shannon index was increased in the oral cavity (Kruskal-Wallis test, $P = 0.0198$) but decreased in the gut ($P = 0.0034$)". Authors should indicate if the p-value is significant. Otherwise it is confusing if the increase/decrease is significant or not.
- The authors should consider to plot the data using Venn diagram showed the shared and unique operational Taxonomic Unit (OTU) all samples from gut, urine, oral and pharynx. Furthermore the authors should consider to plot the distribution of microbial community by using Circos at phylum level to compare gut, urine, oral and pharynx from IgAN patient vs control and early stage vs advanced.
- "In particular, the phylum Firmicutes was significantly decreased in oral cavity,
- pharynx, and gut of IgAN patients contrast to HCs, while there was no distinct difference in urine samples". The authors did not specify if the decrease is significant or not and a p-value was not given.
- "Furthermore, bacteria genera of nine in gut and seven in urine were changed obviously in the advanced stage of IgAN patients compared to early ones ($P < 0.05$, Figure S2). However, there were no significant differences characterized in gut and urine microbiome under much stricter filtering condition". These sentences does not make sense,, Could the authors clarify this?
- "After setting a stricter threshold ($P < 0.001$), the changes of microbiota in the oral and pharynx were much more obvious than that of gut and urine. The authors should clarify the threshold they use for the statistical analysis and when the threshold is statistically significant or not. It is hard to interpret the data.
- The authors studied the correlation of clinical parameters with the differential microbial genera, including eGFR, urea, and complement C3. In the following analysis of the identification and validation of microbial classifiers, the authors include haematuria to refine the model, being crucial. Could the authors use urinalysis data (level of blood, pH, the colour of the urine) to study their correlation with the differential microbial genera?
- Gender differences can be an essential factor not only in human research but also in preclinical research. IgAN Male patients presented with more severe clinical and pathological changes than female patients. The authors should consider analysing the data in gender categories. Clustering the data by gender might another important comparison as early-stage vs advanced stage.

The authors for the first time conducted a large-scale 16S rRNA sequencing in IgAN patients and healthy volunteers across oral, pharynx, gut, and urine samples, revealing the comprehensive niche-specific patterns of microbial dysbiosis in IgAN. The study design is novel in general, the statistical analysis is comprehensive and solid, and the results provide clinical implications. However, I do have several concerns as listed below.

1. For the study design section: 1) the inclusion criteria of healthy controls should be listed (page 27 line 461). Did the participants take any medications? How did the authors deal with medication use in the analysis? 2) the authors used matched case-control study design (page 5 line 98). What kind of matching method is used? 3) The units for the continuous variables in table 1 should be added.
2. For the statistical analysis section: 1) MaAsLin2 was used for multivariable association analysis (page 33, line 568). What variables are included for the analysis? And in what analysis exactly was MaAsLin2 used? It is not clear to me. 2) Multiple comparison issue should be at least discussed. Or maybe for the main analysis, $P < 0.001$ could be a choice.
3. I would suggest conducting case-control comparison as the main analysis while the early-advanced CKD stage comparison as the second analysis, in order to present the data in a clearer way.
4. For the results section: 1) it is in general too wordy and could be condensed. For example, the description of taxonomic comparisons would be focused on genus levels and that on phylum levels could be condensed. 2) Overall, the results are quite descriptive without main points. I would suggest reorganizing them into several hierarchical major lines. 3) in figures 2, 3 and 4, the authors seemed to combine discovery cohort and validation cohort to conduct the analysis (am I correct? how is the batch effect handled then?). The strategies to validate the discovered signals in the validation samples were only used in Figure 5. Please justify the rationale and describe it in the manuscript. 4) hematuria is an indicator of an infection, which is common in nephropathy. How is the prediction ability of hematuria in distinguishing IgAN patients from HCs? If it is quite strong by hematuria itself, I would not include it in the microbial prediction model.

Point-to-point response to reviewers' comments

Response to Reviewer #1

1. All the acronyms should be explained at least once in the main text.

Response: Thank the reviewer for pointing this out. We have reviewed the whole manuscript and added the full name of all the acronyms (Lines 51, 87, 97, 103-104).

2. Alpha diversity was investigated by the authors using Shannon indexes. Some of the common alpha diversity indices are the following: (1) Chao Index, (2) Simpson Index, (3) Shannon Index, (4) ACE Index, and (5) Good's Coverage Index. Could the authors explain why they decided to use Shannon index to study alpha diversity and not the other of the indices? At least authors should mentioned why Shannon index was chosen.

Response: We thank the reviewer for this suggestion. The reason that we reported the Shannon index is that it takes into account both the number of species and the evenness of the community, and has been widely used to evaluate alpha diversity (1). In addition, we have addressed other alpha diversity indices that you mentioned, including Chao Index, Simpson Index, ACE Index, and Good's Coverage Index in Table S2, which showed the tendency of similar alteration as Shannon index. Moreover, the relevant description was added in lines 125-126.

3. "Shannon index was increased in the oral cavity (Kruskal-Wallis test, $P = 0.0198$) but decreased in the gut ($P = 0.0034$)". Authors should indicate if the p-value is significant. Otherwise it is confusing if the increase/decrease is significant or not.

Response: Thank the reviewer for this suggestion. We added the relevant contents to the revised manuscript in lines 127, 128 and 515.

4. The authors should consider to plot the data using Venn diagram showed the shared and unique operational Taxonomic Unit (OTU) all samples from gut, urine, oral and pharynx. Furthermore the authors should consider to plot the distribution of microbial community by using Circos at phylum level to compare gut, urine, oral and pharynx from IgAN patient vs control and early stage vs advanced.

Response: Thank the reviewer for these suggestions. We added the Venn diagrams to the supplementary (Fig. S1) and the relevant contents to the revised manuscript (Lines 121-125). For the Circos diagram, we plotted it to show the compositions

of top five phyla with information of relative abundance, disease status and sample niches as follows (FIG R1). While to some degree, we found the main information that we wanted to highlight, namely the microbial alterations between disease status and general differences among body niches, was not very clear in the Circos diagram. Therefore, we remained the stacked bar chart to show the dominant phyla between IgAN patients and HCs among four body niches (Fig. 2C and Fig. 5A).

FIG R1 The Circos diagram of the phylum level for different stages of IgAN patients and HCs among four niches (oral, pharynx, gut, urine).

5. “In particular, the phylum Firmicutes was significantly decreased in oral cavity, pharynx, and gut of IgAN patients contrast to HCs, while there was no distinct difference in urine samples”. The authors did not specify if the decrease is significant or not and a p-value was not given.

Response: Thank the reviewer for pointing this out. We added the FDR *P*-values

to the revised manuscript in line 146.

6. “Furthermore, bacteria genera of nine in gut and seven in urine were changed obviously in the advanced stage of IgAN patients compared to early ones ($P < 0.05$, Figure S2). However, there were no significant differences characterized in gut and urine microbiome under much stricter filtering condition”. These sentences does not make sense, Could the authors clarify this?

Response: Thank the reviewer for this reminding. We deleted the meaningless sentences of the revised manuscript in line 270.

7. “After setting a stricter threshold ($P < 0.001$), the changes of microbiota in the oral and pharynx were much more obvious than that of gut and urine. The authors should clarify the threshold they use for the statistical analysis and when the threshold is statistically significant or not. It is hard to interpret the data.

Response: Thank the reviewer for this suggestion. We added the relevant contents to the revised manuscript in lines 517-520.

8. The authors studied the correlation of clinical parameters with the differential microbial genera, including eGFR, urea, and complement C3. In the following analysis of the identification and validation of microbial classifiers, the authors include haematuria to refine the model, being crucial. Could the authors use urinalysis data (level of blood, pH, the colour of the urine) to study their correlation with the differential microbial genera?

Response: Thank the reviewer for this question. In this study, the renal function of included healthy controls was judged by creatinine in blood, so routine urine data of HCs were not detected. It's a pity that the urinalysis data (level of blood, pH, the color of the urine) of HCs were not included. Therefore, the correlation analysis with differential microbial genera was not conducted. In future studies, we will consider including the level of blood, pH, and color of the urine for analysis.

9. Gender differences can be an essential factor not only in human research but also in preclinical research. IgAN Male patients presented with more severe clinical and pathological changes than female patients. The authors should consider analyzing the data in gender categories. Clustering the data by gender might

another important comparison as early-stage vs advanced stage.

Response: Thank the reviewer for this suggestion. The purpose of our study is to investigate the changes of microbiota in IgAN among multiple niches, and we also notice that gender may influence the progression of chronic kidney disease (2, 3). However, in our study, the gender ration was relatively balanced (Table 1 and Table S1), and the PERMANOVA (permutational multivariate analysis of variance) test displayed that gender had no significant effect on the microbial community in the early and advanced stages of IgAN patients (P value of PERMANOVA in the oral, pharynx and gut samples were greater than 0.05, indicating that gender had no significant effect on the microbial community). Although the P value of urine samples in PERMANOVA was less than 0.05, the P value of betadisper analysis was also less than 0.05 (Table R1), which indicated that the dispersion of urine microbial data was relatively large and not caused by gender factor. We would consider the influence of gender on development of IgAN in the future.

Table R1 the results of PERMANOVA and betadisper test

Niches	P -value (PERMANOVA)	P -value (betadisper)
Oral	0.225	0.12
Pharynx	0.201	0.548
Gut	0.121	0.2
Urine	0.001	0.007

Note: PERMANOVA, permutational multivariate analysis of variance.

Response to Reviewer #3

The authors for the first time conducted a large-scale 16S rRNA sequencing in IgAN patients and healthy volunteers across oral, pharynx, gut, and urine samples, revealing the comprehensive niche-specific patterns of microbial dysbiosis in IgAN. The study design is novel in general, the statistical analysis in comprehensive and solid, and the results provide clinical implications. However, I do have several concerns as listed below.

1. For the study design section: 1) the inclusion criteria of healthy controls should be listed (page 27 line 461). Did the participants take any medications? How did the authors deal with medication use in the analysis?

Response: Thank the reviewer for comments and nice reminding. We added the

inclusion criteria of healthy controls to the revised manuscript in lines 412-413, 418-419. To minimize the influence of medication use, participants treated with antibiotics, microbial agents, traditional Chinese medicine within 4 weeks, glucocorticoid or other immunosuppressants within 24 weeks before enrollment were excluded (lines 420-422). Thus, these results in our work were not affected by medication use.

2) the authors used matched case-control study design (page 5 line 98). What kind of matching method is used?

Response: Thank the reviewer for this question. We're sorry for the misunderstanding caused by our incorrect description. The patients and controls recruitments were strictly followed the inclusion and exclusion criteria, such as age and BMI distribution (Lines 412-422). Then, statistical analysis showed that age, gender, BMI had no significant differences between IgAN patients and HCs (Table 1, Wilcoxon rank sum test for continuous variables, such as age, BMI; Chi-square test for categorical variables, such as gender). To avoid potential confusion, the term 'matched' was replaced, and relevant description has been modified in the manuscript of lines 96-98.

3) The units for the continuous variables in table 1 should be added.

Response: Thank the reviewer for nice reminding. We added the units for Creatinine, Urea and eGFR in both Table 1 and Table S1 of the revised manuscript.

2. For the statistical analysis section: 1) MaAsLin2 was used for multivariable association analysis (page 33, line 568). What variables are included for the analysis? And in what analysis exactly was MaAsLin2 used? It is not clear to me.

Response: Thank the reviewer for these questions. We added the variables of the MaAsLin2 to the revised manuscript in lines 521-522. Although there were no statistically significant differences in gender, age or BMI between IgAN patients and HCs (Table 1), in order to control any potential bias caused by individual heterogeneity, these variables were still used in MaAsLin2 to conduct multivariate association analysis. The relevant results have been presented in supplementary materials (Fig. S2), which were almost consistent with the results without variable adjustment.

2) Multiple comparison issue should be at least discussed. Or maybe for the main analysis, $P < 0.001$ could be a choice.

Response: Thank the reviewer for reminding. We carefully checked the main analysis (differential phylum/genus analysis, Co-abundance networks analysis, Spearman's rank correlations analysis) and added FDR-corrected P values to the revised manuscript.

3. I would suggest conducting case-control comparison as the main analysis while the early-advanced CKD stage comparison as the second analysis, in order to present the data in a clearer way.

Response: Thank the reviewer for this suggestion, and we have reorganized the manuscript. To make the manuscript more readable, we reported IgAN related microbial alterations, microbial community characteristics and biomarkers as the main results. Then, we further explored the microbial alterations and potential biomarkers between the early and advanced stages of IgAN patients, which were rearranged as a separate part of the revised manuscript (Lines 253-307).

4. For the results section: 1) it is in general too wordy and could be condensed. For example, the description of taxonomic comparisons would be focused on genus levels and that on phylum levels could be condensed.

Response: Thank the reviewer for this suggestion. We streamlined the taxonomic comparisons of phylum level and improved the relevant descriptions to be more concise and readable in the revised manuscript (Lines 142-146, 257-259).

2) Overall, the results are quite descriptive without main points. I would suggest reorganizing them into several hierarchical major lines.

Response: Thank the reviewer for this suggestion. We have reorganized the full text and made improvements to the revised manuscript.

3) in figures 2, 3 and 4, the authors seemed to combine discovery cohort and validation cohort to conduct the analysis (am I correct? how is the batch effect handled then?). The strategies to validate the discovered signals in the validation samples were only used in Figure 5. Please justify the rationale and describe it in the manuscript.

Response: Thank the reviewer for raising this doubt. We're sorry for this

confusion. Actually only discovery cohort data was used in figures 2, 3 and 4, so no batch effects were involved. We added the relevant description on the study design and analyses in the revised manuscript. Details could be found in the INTRODUCTION (Lines 81-82, 84-86), RESULTS (Lines 124-125, 214 etc.) and DISCUSSION (Lines 317-318, 319).

4) hematuria is an indicator of an infection, which is common in nephropathy. How is the prediction ability of hematuria in distinguishing IgAN patients from HCs? If it is quite strong by hematuria itself, I would not include it in the microbial prediction model.

Response: Thank the reviewer for this question. We have found no studies that use hematuria alone to distinguish IgAN patients from HCs currently. In addition, other diseases and external factors, such as pressure or injury, may also cause hematuria, which is not specific for IgAN. Therefore, we deleted the models of combining the microbes and hematuria in the revised manuscript according to your comments.

REFERENCES

1. Sherwin WB, Prat IFN. The Introduction of Entropy and Information Methods to Ecology by Ramon Margalef. Entropy (Basel) 2019;21.<https://www.mdpi.com/1099-4300/21/8/794>.
2. Swartling O, Rydell H, Stendahl M, Segelmark M, Trolle Lagerros Y, Evans M. CKD Progression and Mortality Among Men and Women: A Nationwide Study in Sweden. Am J Kidney Dis 2021;78:190-199 e191. <https://www.sciencedirect.com/science/article/pii/S0272638621000184?via%3Dihub>.
3. Ricardo AC, Yang W, Sha D, Appel LJ, Chen J, Krousel-Wood M, et al. Sex-Related Disparities in CKD Progression. J Am Soc Nephrol 2019;30:137-146. https://journals.lww.com/jasn/Fulltext/2019/01000/Sex_Related_Disparities_in_CKD_Progression.15.aspx.

May 4, 2023

Prof. xueqing Yu
Guangdong Provincial People's Hospital
106th, Zhongshan Road II
Guangzhou
China

Re: Spectrum05202-22R1 (Systematic microbiome dysbiosis are associated with IgA nephropathy)

Dear Prof. xueqing Yu:

Your manuscript has been accepted, and I am forwarding it to the ASM Journals Department for publication. You will be notified when your proofs are ready to be viewed.

Sincerely,

Wei-Hua Chen
Editor, Microbiology Spectrum
